# Convergent antibody evolution and clonotype expansion following influenza virus vaccination

David Forgacs[1☯], Rodrigo B. Abreu[1☯], Giuseppe A. Sautto[1☯], Greg A. Kirchenbaum[1¤], Elliott Drabek[2], Kevin S. Williamson[2], Dongkyoon Kim[2], Daniel E. Emerling[2], Ted M. Ross[1,3]*

1 Center for Vaccines and Immunology, University of Georgia, Athens, GA, United States of America,
2 Atreca, Inc., South San Francisco, CA, United States of America, 3 Department of Infectious Diseases, University of Georgia, Athens, GA, United States of America

☯ These authors contributed equally to this work.
¤ Current address: Cellular Technology Ltd., Shaker Heights, OH, United States of America
* tedross@uga.edu

**Data Availability Statement:** All relevant data are within the manuscript and its Supporting Information files.

## Abstract

Recent advances in high-throughput single cell sequencing have opened up new avenues into the investigation of B cell receptor (BCR) repertoires. In this study, PBMCs were collected from 17 human participants vaccinated with the split-inactivated influenza virus vaccine during the 2016–2017 influenza season. A combination of Immune Repertoire Capture (IRC™) technology and IgG sequencing was performed on ~7,800 plasmablast (PB) cells and preferential IgG heavy-light chain pairings were investigated. In some participants, a single expanded clonotype accounted for ~22% of their PB BCR repertoire. Approximately 60% (10/17) of participants experienced convergent evolution, possessing public PBs that were elicited independently in multiple participants. Binding profiles of one private and three public PBs confirmed they were all subtype-specific, cross-reactive hemagglutinin (HA) head-directed antibodies. Collectively, this high-resolution antibody repertoire analysis demonstrated the impact evolution can have on BCRs in response to influenza virus vaccination, which can guide future universal influenza prophylactic approaches.

## Introduction

Influenza virus is a highly contagious viral respiratory disease that causes hundreds of thousands of deaths and millions of hospitalizations every year [1]. Annual vaccination is recommended to reduce influenza virus disease severity and limit transmission of the virus by eliciting antibodies that primarily target the hemagglutinin (HA) glycoprotein [2]. The antibody response elicited by the current seasonal influenza vaccine is predominantly strain-specific and highly dependent on pre-existing immunity and imprinting [3, 4]. However, how efficiently influenza virus vaccination recalls different pre-existing B cell memory responses in different participants and how they are correlated with protection needs further detailed

**Funding:** This study was supported by the NIH contract HHSN272201400004C (NIAID Centers of Excellence for Influenza Research and Surveillance, CEIRS), University of Georgia (US), UGA-001 and the Georgia Research Alliance (GRA-001). The content is solely the responsibility of the authors and does not necessarily represent the official views of the NIH. The funder provided support in the form of salaries for authors [DF, RBA, GAS, GAK], but did not have any additional role in the study design, data collection and analysis, decision to publish, or preparation of the manuscript. The specific roles of these authors are articulated in the 'Author contributions' section. Some of the authors are affiliated with Atreca, Inc., however, they provided no funding for the study, only assisted with the data collection and the preparation of the manuscript.

**Competing interests:** Some of the authors are affiliated with Atreca, Inc., but the company provided no funding for the study and no competing interests exist. The commercial affiliation of those authors does not alter our adherence to PLoS ONE policies on sharing data and materials.

investigation. Circulating viral isolates facilitate escape from pre-existing antibodies by continuous antigenic drift [5], which is the main reason why frequent reformulation of the seasonal influenza virus vaccine is required. Therefore, there is a need to identify broadly reactive neutralizing antibodies that are effective against various influenza viral strains as a means of developing more effective immunotherapeutic strategies and tools to help in designing broadly prophylactic countermeasures. One way to discover such antibodies is through B cell receptor (BCR) repertoire analysis that allows for a meticulous investigation of the B cell-mediated immune response. Antigen-specific BCR repertoire analysis has been reported in connection with infectious and autoimmune diseases, including HIV [6], celiac disease [7], and influenza [3, 8–10]. In this context, it is important to define the BCR repertoire in a large cohort of patients belonging to different age groups as a means of identifying potential antibody signatures following influenza vaccination. This strategy opens up new possibilities for a more robust and detailed B cell repertoire analysis, and for the reliable evaluation of the divergences and convergences in antibody sequences following influenza vaccination.

Importantly, the antigen specificity of individual BCRs is largely determined by the complementarity determine regions (CDR) encoded within the variable V(D)J gene segments of the Ig heavy and light chains. Of greatest importance for antigen binding to BCRs is the CDR3 region, since it is located at the very tip of the BCR [11]. It spans the 3' end of the V gene, the entire D gene in the case of the heavy chain, and the 5' end of the J gene segment. The importance of the amino acids encoded in the CDR3 region is further highlighted by the fact that this region is a hot spot for non-synonymous, affinity enhancing mutations [11, 12]. Moreover, numerous co-crystal structures of antibody/antigen complexes further highlight the critical importance of CDR3-encoded amino acid residues to antigen binding [13, 14].

It is essential to maintain a highly diverse BCR repertoire so that individual B cells can be activated upon encountering their cognate antigen to ensure that an antibody response can be mounted against a myriad of potential targets. To that end, immature B cells migrate to the periphery to complete their development following rearrangement of V(D)J gene segments in the bone marrow [15, 16]. B cell clones that acquire affinity enhancing mutations in their BCR through the process of somatic hypermutation (SHM) will exhibit improved antigen-binding capability and therefore will receive preferential T cell help as antigen concentrations decline during vaccination or an acute infection [17]. In addition to seeding the memory B cell compartment, these affinity matured B cells also differentiate into antibody-secreting plasmablasts (PB) that serve to initiate the increase in systemic antibody levels in response to an offending antigen. This process is termed affinity maturation and ensures that a high-affinity antibody response is generated following antigen encounter. Moreover, in the event antibody titers decline or are insufficient to prevent re-infection, such as in the case of seasonal influenza viruses that have undergone antigenic drift, these affinity matured memory B cells generated in the primary response will be poised to rapidly respond upon secondary antigen encounter. Collectively, through the concerted contributions of V(D)J recombination and SHM within germinal centers, these processes increase the likelihood that a healthy individual will mount a successful antibody response to a multitude of antigens, including those originating from hypervariable pathogens such as the influenza virus.

Given the importance of the PB response in providing a glimpse into the immediate reaction of the immune system to vaccination [16], in this study, 7,777 PBs from 17 vaccine recipients were subjected to BCR repertoire analysis. PBMCs were collected at the presumed peak of PB expansion, 7–9 days after immunization with the 2016–2017 Northern hemisphere seasonal split-inactivated influenza virus vaccine. Following paired heavy-light chain single cell sequencing, preferential chain pairing, expanded clonotypes, and public antibodies which are

a product of convergent evolution between PB BCR repertoires in multiple participants were assessed with a special emphasis on the heavy chain CDR3 regions (HCDR3).

Overall, the goal was to examine clonotype expansion and convergent evolution that are indicative of the elicitation of antibody signatures specific to the influenza virus vaccination. When multiple individuals have similar PB BCR repertoires elicited by the vaccination independently, those sequences are indicative of convergent evolution between participant PB antibodies and represent a convergence group that emerged in reaction to the influenza vaccine antigens. In order to better understand the functional significance of convergent PB BCRs, four PB-derived clonotypes were selected from the IgG sequence repertoires and recombinantly expressed (S1 Graphical abstract). Three of these were public antibodies from clonotypes that were found to have independently arose in at least two of the participants in this study, and one private antibody was selected for having a previously not well described IGHV1-69-2 heavy chain variable segment. This participant selection and antibody discovery pipeline successfully identified both public and private antibodies that bound the HA head region, and confirmed that PBs were a preferential compartment enriched in antigen-specific BCRs as a direct result of the influenza virus vaccination.

## Material and methods

### Participant selection

Eligible participants between the ages of 17 and 65 years old were enrolled between September 2016 and March 2017 at the UGA Clinical and Translational Research Unit (CTRU) in Athens, Georgia, USA (S1 Table). Participants were recruited and enrolled with written, informed consent. The study procedures, informed consent, and data collection documents were reviewed and approved by the Institutional Review Board of the University of Georgia. Exclusion criteria included documented contraindications to Guillain-Barré syndrome, dementia or Alzheimer's disease, allergies to eggs or egg products, estimated life expectancy less than 2 years, immunocompromising condition, or concurrent participation in another influenza vaccine research study. During the course of their first visit, the participants had their blood drawn and were subsequently vaccinated with the standard dose (15 μg/antigen) of the split-virion (IIV) quadrivalent seasonal influenza vaccine Fluzone (Sanofi Pasteur). The vaccine consisted of 4 influenza strains in accordance with the recommendations of the WHO for the 2016–2017 Northern hemisphere influenza season: A/California/7/2009 (H1N1), A/Hong Kong/4801/2014 (H3N2), B/Phuket/3073/2013 (Yamagata-lineage), and B/Brisbane/60/2008 (Victoria-lineage). 7–9 days later, around the time of peak PB response, the participants returned and their blood was drawn which was used for PBMC sorting and sequencing [18, 19]. Participants were not monitored for influenza virus infection due to low levels of circulation in the community during that time period, but participants were asked during each visit if they had experienced flu-like symptoms, and those who did were excluded from the study. 148 eligible participants were enrolled, of which 17 (10 females, 7 males) were selected who had not received a seasonal influenza vaccine in the 3 years prior. The funding source had no role in sample collection, nor decision to submit the manuscript for publication.

### Monoclonal antibodies (mAbs)

Heavy and light chains of IgG1, λ public (#3978, #5589 and #7665) and private (#1664) mAbs were synthesized and cloned in the pcDNA3.4 vector (Thermo Fisher Scientific) and expressed in Human Embryonic Kidney (HEK) 293F cells by GenScript (Piscataway, NJ, USA). Expressed mAbs were quantified by ELISA from conditioned media harvested from transfected cells and their binding and functional activity was evaluated as described below.

## Serological assays

Enzyme-linked immunosorbent assays (ELISAs) were performed both on the human poly-clonal and mAbs, the latter expressed in this study following the procedures previously utilized in our laboratory [20–22]. Briefly, plates were coated with 50 ng/well of recombinant hemag-glutinnin (rHA) [23, 24], blocked, antibodies were serially diluted starting from a 1:500 dilu-tion for polyclonal sera, and from a concentration of 20 μg/mL for the mAbs. HA-specific IgG antibodies were detected, absorbance measured at 414 nm, and HA-specific IgG equivalent concentrations were calculated based on an 8-point standard curve generated using a human IgG reference protein (Athens Research and Technology, Athens, GA, USA).

Hemagglutination inhibition assays (HAIs) were also performed to measure functional antibodies capable of inhibiting virus agglutination in accordance with previously published methods [20–22, 25]. In short, serially-diluted receptor destroying enzyme (RDE) treated sera or mAbs starting from an initial dilution of 1:10 or 20 μg/mL, respectively, were incubated with an equal volume of influenza virus. Erythrocytes were added and the HAI titer was deter-mined by the reciprocal dilution of the last well that contained non-agglutinated RBCs.

## Cellular assays

Enzyme-linked immune absorbent spot (ELISpot) assays were performed as previously described [20]. Plates were read using the S6 macro ELISpot reader and automated counting was performed using ImmunoSpot (v5.1) software (Cellular Technology Limited, Shaker Heights, OH, USA).

PBMCs were subjected to *in vitro* differentiation to stimulate memory B cells by incubating with 500 ng/mL R848 (Invivogen, San Diego, CA, USA) and 5 ng/mL rIL-2 (R&D, Minneapolis, MN, USA) for 7–9 days at 37°C in 5% $CO_2$ as previously described [20]. Total and rHA-specific IgG abundance of the conditioned medium supernatants were assessed by ELISA starting at a 1:5 dilution, and the frequency of B cells was assessed by CD19 surface labeling and flow cytometry.

For flow cytometry analysis, human PBMCs (~5 x $10^6$ live cells) were treated as previously described [20]. In brief, PBMCs were first treated with Fc receptor blocking solution (Biolegend, Dedham, MA, USA) then stained for 30 min on ice using titrated quantities of fluorescently con-jugated mAbs. After completion of surface labeling, cells were washed extensively with staining buffer prior to fixation with 1.6% paraformaldehyde in staining buffer for 15 min at RT. Follow-ing fixation, cells were pelleted by centrifugation at 400x*g* for 5 min, resuspended in staining buffer and maintained at 4°C protected from light until acquisition. Data acquisition was per-formed using the BD FACSARIA Fusion and analysis performed using FlowJo (FlowJo LLC, Ashland, OR, USA). Compensation values were established prior to acquisition using appropri-ate single stain controls. PBs were defined as CD3/CD14$^{neg}$ CD19$^+$, CD27$^+$, CD38$^{++}$ cells as pre-viously described [26, 27]. Flow cytometric analysis and cell sorting were performed on BD FACSAria Fusion using the BD FACSDiva Software. IgG-enriched PBs were isolated by gating for CD19$^+$CD20$^{low/−}$CD27$^+$CD38$^{high}$IgA$^-$IgM$^-$IgD$^-$CD3$^-$CD14$^-$ live cells. The PBs were single-cell sorted into 384-well PCR plates containing lysis buffer (10 mM Tris-HCl, pH 8.0), 2 mM dNTPs (New England Biolabs), 2 mM DTT (Sigma-Aldrich), 0.2 mg/mL BSA (New England Biolabs), 0.01% IGEPAL-630 (Sigma-Aldrich) and 0.5 unit/μl of RiboLock RNase Inhibitor (ThermoFisher Scientific). Single-cell sorted plates were stored at −80°C until use for paired Ig heavy and light chain sequencing using Atreca's Immune Repertoire Capture® technology.

## IgG sequencing

Reverse transcription, PCR, barcode assignment, sequence assembly, V(D)J assignment, and identification of mutations were performed on the single-cell sorted samples as previously

described [28, 29] with the following modifications: desthiobiotinylated oligo(dT) was used for reverse transcription, cDNA was extracted using Streptavidin C1 MyOne beads (Life Technologies), DNA library concentrations were determined using qPCR (KAPA SYBR®FAST qPCR Kit for Illumina, Kapa Biosystems), and a minimum coverage of 10 reads was required from each chain assembly to be included in the sequence repertoires. V(D)J assignment and mutation identification were performed using the International Immunogenetics Information System (IMGT, http://www.imgt.org). All heavy and light chain sequences were independently gapped using IMGT's HighV-QUEST 1.7.0 [30–32] and deposited in the NCBI GenBank database as a Targeted Locus Study project (heavy chain: KEOV00000000, light chain: KEOU00000000). Paired heavy and light chain sequences were concatenated by introducing a number of gaps after the heavy chain sequence that ensured that when aligned, all light chain sequences start congruously at the same nucleotide position.

## Heavy-light chain usage

Column charts representing IgG subclass usage, heavy and light chain kappa/lambda usage, and heavy chain variable region usage were plotted in GraphPad Prism 8.0. A Circos plot was created for the depiction of heavy-light chain pairings [33]. PBs were sorted by heavy and light chain variable region subgroups, in a way that the left hemisphere of the Circos plot represents the heavy chain (IGHV) groups, and the right hemisphere represents the light chain kappa and lambda (IGKV/IGLV) groups proportionally. Each PB is equally weighted and links demonstrate the pairing between the heavy and the light chain components of the individual PBs. Preferential joining of V(D)J and VJ gene segments for heavy and light chains was performed using the IMGT/HighV-QUEST and IMGT/StatClonotype tools freely available on the IMGT website. Nucleotide sequences for heavy and light chain variable portions were uploaded to the IMGT/HighV-QUEST platform and statistical analysis was performed using the IMGT/StatClonotype tool.

## Pie chart and phylogenetic tree depiction of clonotypes

All heavy and light chain sequences from each participant were independently aligned and arranged by phylogenetic order by creating a neighbor joining guide tree with 100 bootstraps using the Tamura-Nei genetic distance model. Subsequently, a list of sequences with unique heavy/light chain IMGT identifiers was created, sequences with the same name (referred to throughout as a clonotype) were collapsed into a single entry, and depicted as a pie chart using GraphPad Prism 8.0. The heavy chain variable segment (IGHV) of the 3 largest segments were reviewed (S2 Table), and in cases where at least two of them belonged to the same IGHV subgroup (e.g. IGHV1), a tree was created to illustrate the location and the relative distance between these highly expanded clonotypes. For these phylogenetic trees, maximum likelihood trees using the GTR+GAMMA+I nucleotide model with 100 bootstraps were created using RAxML [34]. The 3 largest pie slices and the corresponding clades on the tree were colored orange for the largest, blue for the second largest, and brown for the third largest slice, as long as they all belonged to the same IGHV subgroup.

## Public B cell sequence identity heat maps

Convergent evolution was assessed by finding groups of PBs whose heavy chain variable region V and J segments were identical, their nucleotide pairwise identity including the D segment was over 80%, and HCDR3 peptide sequence pairwise identity was over 75% compared to at least one other PB from a different participant in the same group [35, 36]. To be considered as a public antibody group, the groups also had to have a minimum of 10 cells, with an HCDR3

greater than 15 amino acids in length. While the HCDR3 peptide sequence pairwise identity cut-off was set at 75%, there were only two B cells whose identities were below 85% (#7031 in pubCDR3-1 and #7739 in pubCDR3-4). However, as they both had identical V and J segments and heavy chain nucleotide pairwise identities above the cut-off (83% and 93% respectively), we felt it was appropriate to include them in the analysis. Heat maps for the four resulting convergence groups (pubCDR3-1 through pubCDR3-4) were created using GraphPad Prism 8.0 to illustrate the similarity between sequences that belonged to the same pubCDR3 group. The four panels represent heavy and light chain variable region nucleotide sequences and CDR3 peptide identities. The plotting order of the B cells are conserved between panels; blocks of B cells from different participants are separated by lines.

## Results

### Serological profiling of participants

In order to select participants with prominent immunological responses to influenza virus vaccination, influenza-specific serological and cellular responses prior to, 7 and 28 days post-vaccination were systematically profiled. Overall, 17 participants were selected for this study who were between 17 and 65 years of age. Each of these participants demonstrated a significant increase in their HAI titers against a panel of historical influenza virus strains indicative of back-boost responses (Fig 1A, S1 Fig). Furthermore, all participants had a significant increase in serological IgG antibodies and HAI titers against all four 2016–2017 vaccine strains (Fig 1B, S2 Fig). Memory-derived PB expansion generally occurs 7–9 days post-vaccination. As expected, all 17 participants had a marked PB population defined as CD27[+]/CD38[++] B cells [26, 27] in peripheral blood 7 days post-vaccination. Furthermore, when stained with recombinant HA (rHA) probes, 15–20% of these PBs were influenza A vaccine strain HA specific (Fig 1C). Because probe staining can sometimes be a result of unspecific cell binding [37], we further enumerated the number of antibody secreting cells (ASC) against each of the four vaccine strains in the peripheral blood 7 days post-vaccination by ELISpot (Fig 1D). All selected participants had at least $10^4$ ASC/$10^7$ PBMC, and an average of 13% vaccine-specific ASC (S1 Table). Finally, to evaluate the impact of influenza virus vaccination on the memory B cell compartment, we differentiated memory B cells *in vitro* and quantified the levels of memory-derived antibodies against the four vaccine strains [20]. All 17 participants had an increased antigen-specific antibody response 28 days post-vaccination, indicative of vaccine-specific memory B cell expansion (Fig 1E, S2 Fig).

### Class usage reveals preferential pairing of chains

Following the sequencing of heavy and light Ig genes, we determined that 89.5% of the 7,777 BCRs had an IgG1 constant region subtype (SD = 6.5% between participants), 5.0% encoded for IgG2 and 5.2% for IgG3 (SD = 3.8% and 4.6% respectively), while only 0.3% were IgG4 (Fig 2A). Of the 1,029 clonally expanded lineages, only 8.3% (n = 85) contained PBs that expressed more than one IgG isotype. The most commonly identified isotype switching is between IgG1 and IgG3 (n = 54), followed by IgG1 and IgG2 (n = 26), while it rarely occurs between IgG1 and IgG4 (n = 2) and IgG2 and IgG3 (n = 3). Overall, 41.9% of the light chains were kappa and 58.1% were lambda (SD = 14.8%). The lambda chains were primarily of the λ7 subtype (Fig 2B). Heavy chain variable gene usage across the PB cell population was fairly consistent between participants with preferential usage of $V_H3$ (49.49%±10.2) followed by $V_H1$ (23.25% ±9.0) and $V_H4$ (18.26%±10.3) segments, while $V_H6$ and $V_H7$ were exceedingly rare (0.31% and 0.19% respectively) (Fig 2C). Breaking down light chain class preference by heavy chain variable segment usage, $V_H1$ and $V_H3$ through $V_H5$ all paired with similar frequency of κ:λ light

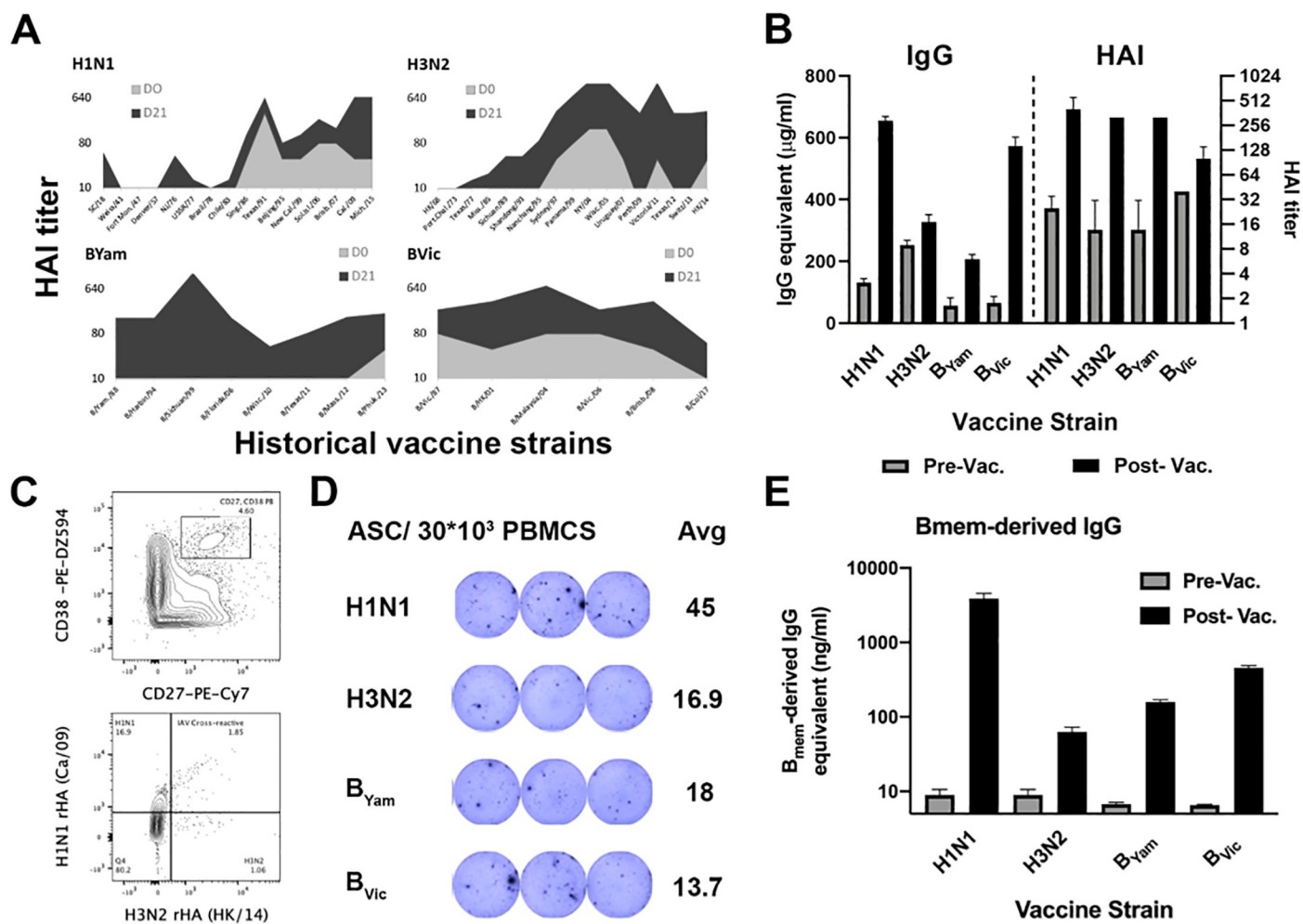

**Fig 1. Serological landscape of D#103, a representative participant in this study.** (A) HAI titer against historical influenza virus strains. (B) Serological IgG antibodies and HAI titer against the 2016–2017 influenza vaccine strains. All samples were run in triplicate. (C) Cell sorting for influenza-specific PBs. (D) Number of antibody secreting cells against the four influenza vaccine strains. (E) Impact of influenza vaccination on the memory B cell compartment.

chains (55.7%-66.1% lambda, 33.9%-44.3% kappa) (Fig 2D). In contrast, 83.3% of $V_H6$ light chains paired with kappa chains, while 72.7% of $V_H2$ and 86.7% of $V_H7$ light chains paired with lambda chains (Fig 2D). Approximately 50% of heavy chains were IGHV3 and they paired most often with IGLV3 (Fig 2E). However, based on the relative frequency of light chains that pair with IGHV3, the pairing to IGLV1 was the most over-represented (58.9%) and IGLV4 was the most under-represented (13.3%) (Fig 2E). The largest divergence from the expected pairing frequencies amongst all heavy-light chains based on $\chi^2$ analysis was between IGHV1 and IGLV4, a pairing that was the most overrepresented in our cohort (65.86% of IGLV4 paired with IGHV1), and between IGHV1 and IGLV1, which was the most underrepresented (11.10% of IGLV1 paired with IGHV1) (Fig 2F).

We also observed a differential usage for specific chains, with some individuals over- or underrepresenting certain variable segments (S3A and S3C Fig) compared to the mean usage amongst all participants. While the number of participants was low, this analysis provides a glimpse into years when chain usage stayed close to the mean (1963–1969 and 1973–1991 for heavy chain variable segments) representing convergent BCR usage, in contrast to other times

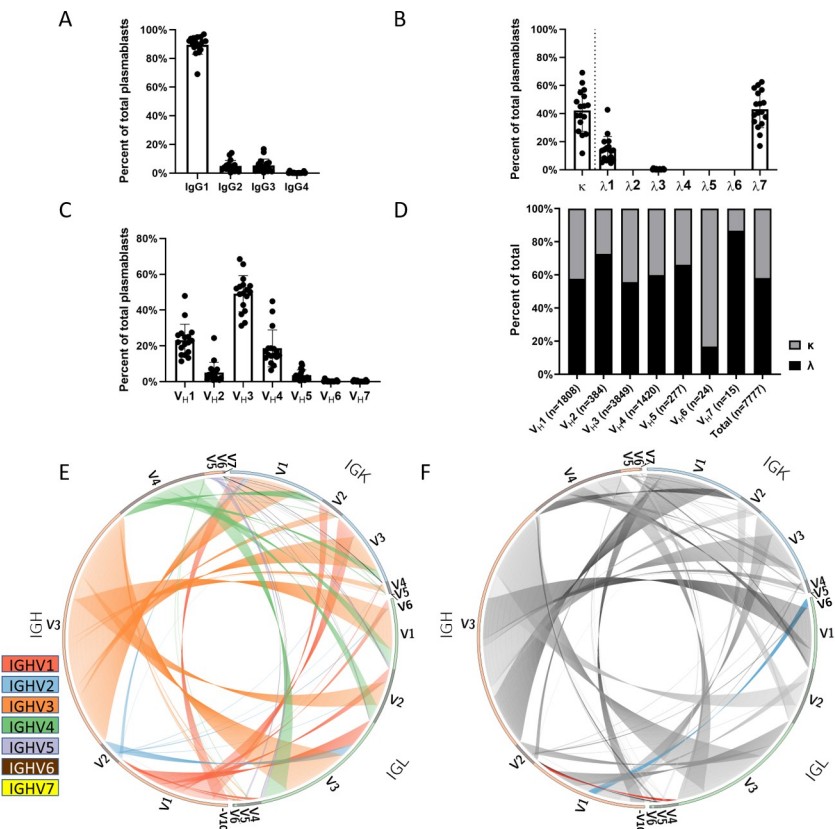

**Fig 2. Chain usage and variable region pairing.** (A) Heavy and (B) light chain constant region subclass usage amongst the IgG PBs. (C) Variable segment usage of participants in this study. (D) Kappa:lambda light chain usage for each heavy chain variable segment. (E-F) Circos plots representing the pairing between heavy and light chain variable regions in the left and right hemisphere respectively. Different colors represent different heavy chain variable segments in (D), while blue represents the most underrepresented (IGHV1/IGLV1) and red represents the most overrepresented (IGHV1/IGLV4) segment pairing in (E).

when the divergence was greater due to an increased evolutionary pressure to innovate [38, 39] (S3B and S3D Fig).

## Clonotype landscapes vary between participants

PB BCR clonotypes for each participant were categorized based on their degree of clonotype expansion, measured by the frequency of heavy-light chain pairing. The top three most highly expanded clonotypes used for each participant (the three largest slices of the pie) were examined (Fig 3, S2 Table). Overall, the proportion of unique clonotypes did not significantly decrease with age (S4 Fig). For participants where at least two of the top three clonotypes belonged to the same IGHV group (15/17 participants), the phylogenetic relationship of the highly expanded clonotypes related to one another within the IGHV group was further assessed (Fig 3B, S2 Table). On average, 10.2% of the participant PBs belonged to the most expanded clonotype (range 3.4% in D#099 to 22.2% in D#103), and 75% of their BCR repertoire was composed of 112 ± 36 unique clonotypes (min: 64 in D#030, max: 195 in D#099). In 4/17 participants, the most expanded clonotypes that belonged to the same IGHV group were identified as closely related, often overlapping sister clades that have likely originated from the same germline sequence (D#038, D#089, D#103, D#122), while in others, the most expanded

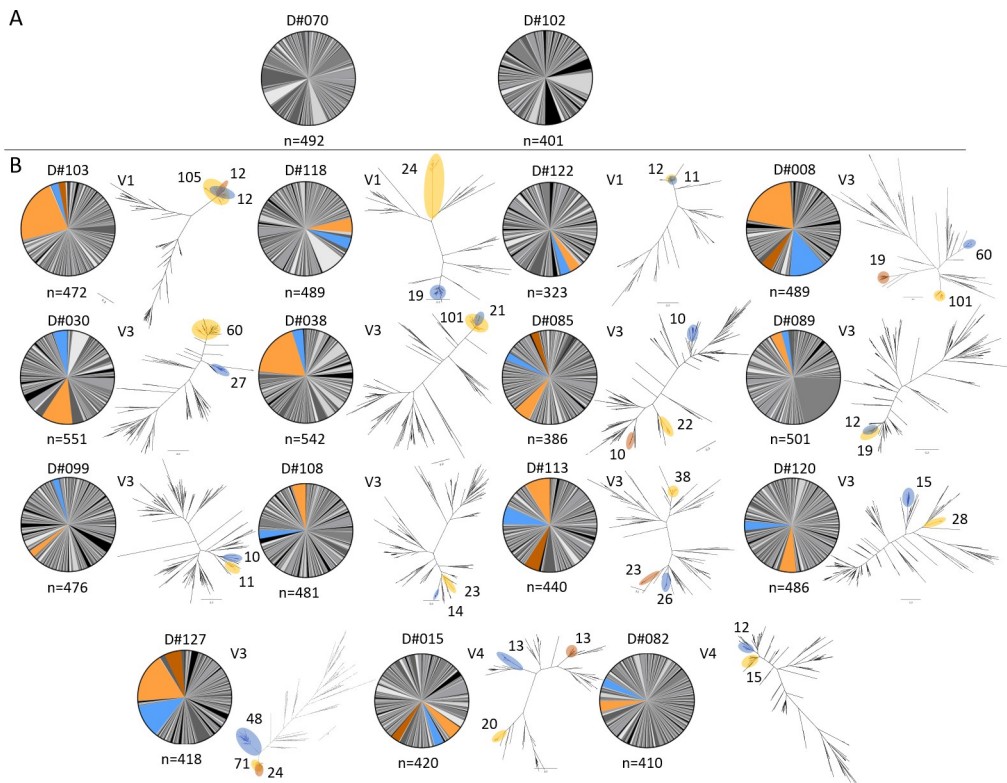

**Fig 3. Clonotype landscape and expanded clonotypes for each participant.** (A) Participants whose three most expanded clonotypes belonged to three different IGHV groups. (B) A phylogenetic tree was included for participants when at least two of the three most expanded clonotypes were of the same IGHV group. This tree only includes those of the top 3 most expanded clonotypes that belong to the same IGHV group. The largest such compartments are shown in orange, the second largest in blue, and the third largest (for participants whose three largest compartments were all the same IGHV group) are shown in brown. On occasion, the most expanded clonotype is not shaded in the pie chart, and is not included on the phylogenetic tree as it represents a different IGHV group than the second and third most expanded clonotypes (*e.g.* D#089). The number of individual PBs are shown next to the highlighted clades on the phylogenetic trees.

clonotypes originated from phylogenetically distinct germline sequences (*e.g.* D#008, D#015, D#085). In some cases, the most expanded clonotype belonged to a different IGHV group than the second and the third, and thus were not included in the phylogenetic tree (*e.g.* D#089, D#118).

## Heavy/light chain V(D)J preferential joining and CDR3 sequence length

Following the analysis of all the heavy and light chain variable regions, there was a consistent preferential association between certain germline genes. For the heavy chain gene sequences, the IGHV3-23 gene segment was predominantly associated with the J4 segment (S5A Fig). The same J4 segment was predominantly associated with the D2-8 gene segment (S5B Fig). For the VD joining, there was an association between the IGHV2-5 segment and the D2-21 gene segment (S5C Fig). For the λ gene sequences, there was an association of the IGLV3-21 gene segment with the J1 segment (S6A Fig). The same J1 segment was predominantly associated with the VK1-5 gene segment (S6B Fig).

The majority of sequenced PBs had an HCDR3 length of 17 amino acids (S7A Fig). In contrast, 9- and 11- amino acid-long LCDR3 regions were most commonly observed for kappa

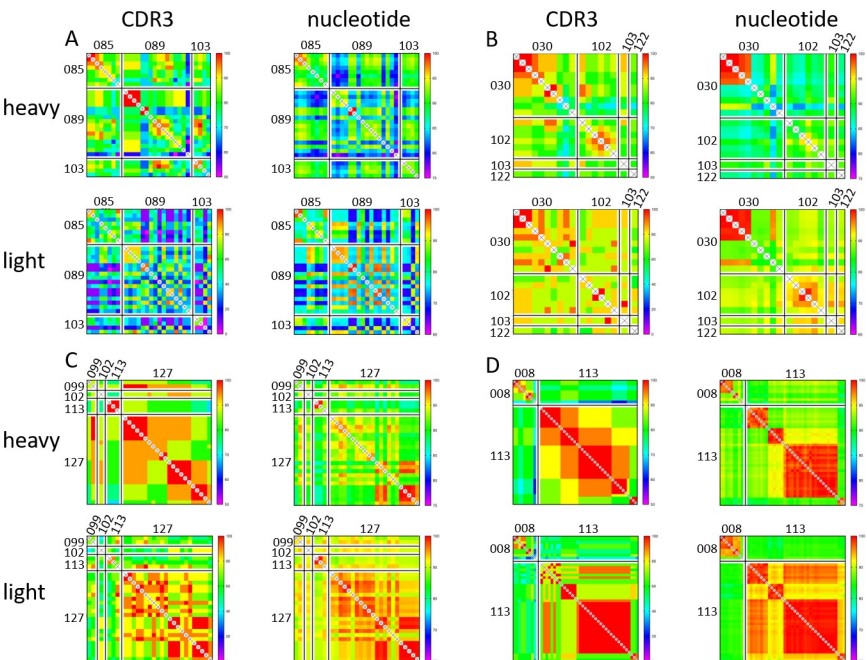

**Fig 4. Pairwise distance heatmaps for all PBs in the four convergent groups.** For each panel, the two heat maps on the left show CDR3 peptide identity for the heavy and the light chain, and the two heat maps on the right show variable region nucleotide identity for the heavy and light chain. (A) Pairwise identity matrix for the pubCDR3-1 group. (B) Pairwise identity matrix for the pubCDR3-2 group. (C) Pairwise identity matrix for the pubCDR3-3 group. (D) Pairwise identity matrix for the pubCDR3-4 group.

and lambda light chains respectively (S7B and S7C Fig). For antibodies exhibiting the most frequent HCDR3 lengths, the corresponding amino acid residue frequencies at each CDR3 position are also shown in the S7 Fig.

## Most participants show signs of convergent antibody evolution

The presence of convergent evolution was determined by assessing the heavy-light chain variable region identity, and more specifically the HCDR3 identity between the PB BCR repertoire of different participants. In 10 out of the 17 participants in our cohort, BCR repertoire sequencing yielded at least one immunoglobulin that demonstrated convergent evolution with a PB from another participant, forming four distinct convergent groups (pubCDR3-1 through pubCDR3-4) with a high likelihood of convergence across participants (Figs 4 and 5, S3 Table). Each of these four groups share a higher percent identity with the members of their pubCDR3 group than to members of the same heavy chain variable segment outside the convergence group (Fig 4, S8 Fig). This result demonstrated that these public PBs are similar in both heavy/light chain variable segment nucleotide identity (S8A Fig) and HCDR3 region identity (S8B Fig).

The pubCDR3-1 group comprised 28 PBs from 3 participants (D#085, n = 8; D#089, n = 16; D#103, n = 4), including the oldest (D#089, born in 1960) and two of the youngest participants (D#103, born in 1997 and D#085, born in 1998) (Fig 4A and S1 Table). The constant regions were primarily IgG1 (85.7%) and λ7 (82.1%), while the remainder were IgG3 and λ1. Both IgG1 and IgG3 were present in D#089 and D#103, while both λ7 and λ1 were present in D#085 and D#103. They all had PBs derived from the IGHV3-23 and IGLV1 or IGLV2

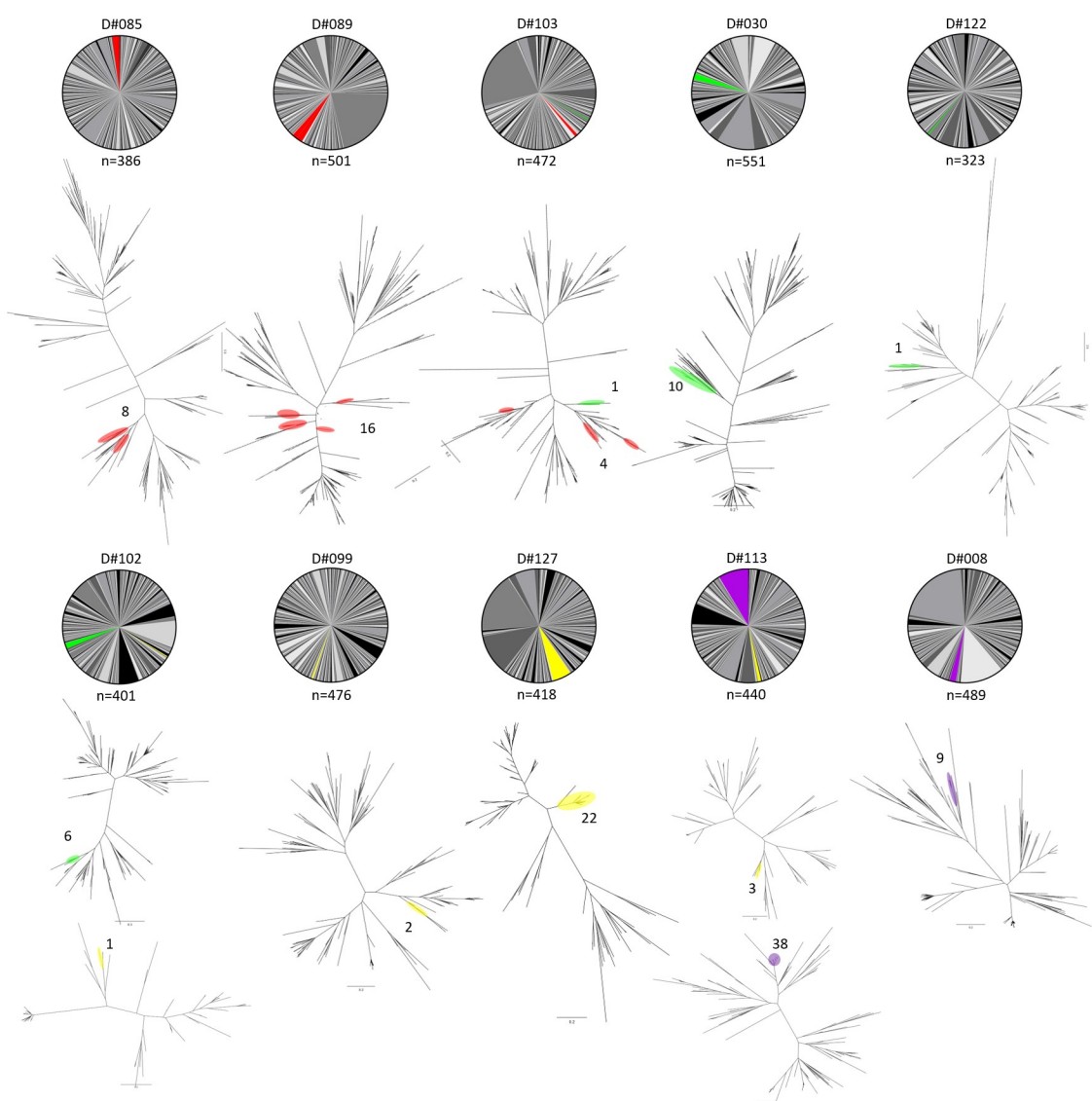

**Fig 5. Clonotype landscape for each participant displaying public BCRs.** Only the 10 participants who had at least one PB belonging to one of the four convergent pubCDR3 groups are shown. The number of individual PBs expressing a public clonotype is shown next to the highlighted clades on the phylogenetic trees.

variable gene subfamilies. D#085 had only IGLV1-derived sequence segments, while the other two participants had both (D#089: 62.5% IGLV1, 37.5% IGLV2; D#103: 75% IGLV1, 25% IGLV2). Even though the heavy chains have passed the selection criteria for a convergent group (nucleotide pairwise identity >80%, HCDR3 amino acid pairwise identity >75%, see Materials and Methods section), the light chains they paired with were highly divergent, even those from the same participants and within the same subfamily. This is due to the selection criteria being based exclusively on heavy chain similarities; several PBs with highly similar heavy chains were paired with a variety of different light chains, particularly when the pubCDR3-1 PBs of D#085 and D#089 CDR3s are compared (Fig 4A). The HCDR3 length of PBs in this group ranged from 20 to 24 amino acids, longer than the most common HCDR3 length in the repertoire dataset (17 amino acids) (S7A Fig).

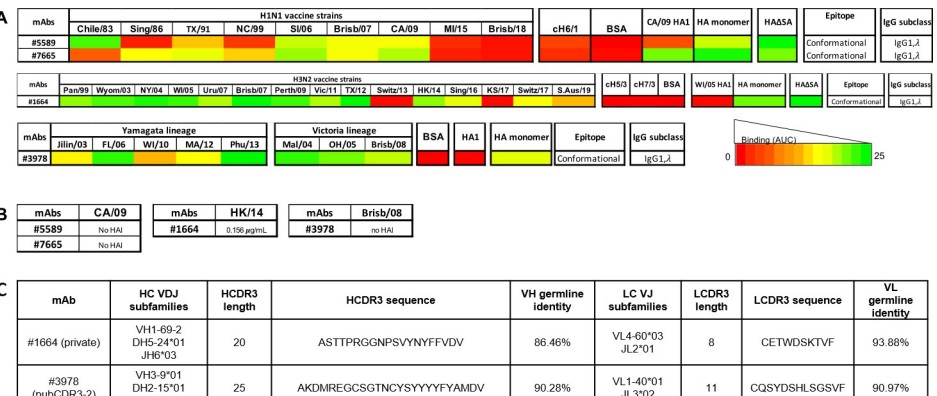

**Fig 6. Expression profile for the four selected mAbs.** (A) Synopsis of H1-, H3-, and IBV-specific human mAbs binding against H1N1, H3N2 and IBV rHA. Binding of mAbs was evaluated against a panel of H1, H3 and IBV rHAs from historical seasonal, pandemic (CA/09) and post-pandemic (Mich/15 and Brisb/18) strains. Binding activity was also tested against COBRA H1N1 P1, X3 and X6 and H3N2 T10 and T11 rHA, the chimeric cH6/1, cH5/3 and cH7/3 rHAs, a truncated HA1 rHA and an rHA monomer. For H1-, H3- and IBV-specific mAbs the HA1 and HA monomer from CA/09, WI/05 and MA/12 were used, respectively. All the mAbs were tested at different 3-fold dilutions, starting from 20 μg/mL. The area under the curve (AUC) was calculated from each dilution curve and reported on the heatmap table. (B) HAI activity of mAbs against the corresponding H1 (CA/09), H3 (HK/14) and IBV (Brisb/08) vaccine strains. All mAbs were tested across a 2-fold serial dilution series, starting from 10 μg/mL. For each mAb, the minimal concentration needed to inhibit hemagglutination is reported in the corresponding tables and expressed in microgram per milliliter. (C) Sequence analysis of heavy and light chain private and public mAbs.

The pubCDR3-2 group comprised 18 PBs from 4 participants (D#030, n = 10; D#102, n = 6; D#103, n = 1; D#122, n = 1) born between 1969 and 1997 (Fig 4B). The constant regions were exclusively IgG1 and predominantly λ7 (94.4%), with a single remaining PB having λ3. They all shared the same variable segments IGHV3-9 and IGLV1-40. These PBs had 26 amino acid long HCDR3 regions, which was the longest of any of the four convergent groups.

The pubCDR3-3 group comprised 28 PBs from 4 participants (D#099, n = 2; D#102, n = 1; D#113, n = 3; D#127, n = 22) born between 1966 and 1995 (Fig 4C). The constant regions were almost entirely IgG1 (96.4%) and paired with λ1 (74.1%) or λ7 (25.9%) light chains. One additional IgG3 sequence belonging to the pubCDR3-3 group paired with a λ7 light chain. They all had IGHV1-2 and IGLV3-1 variable segments, with a 16 amino acid long HCDR3 region.

The pubCDR3-4 group comprised 24 BCRs from 2 participants of similar ages (D#008, n = 9; D#113, n = 38), born in 1966 and 1970 (Fig 4D). All heavy chain constant regions were IgG1, while light chain constant regions were split between λ1 (97.9%) and λ7 (2.1%). They all shared the same variable segments IGHV3-7 and IGLV3-21, with an HCDR3 region composed of 20 amino acids.

## Analysis of selected private and public antibodies

Four representative IgG1, λ antibodies were selected, expressed, and purified for further characterization (Fig 6). Monoclonal antibody (mAb) #1664 was a representative sequence from an IGHV1-69-2 subgroup from D#103 (IGHV1-69-2/D5-24/J6/IGLV4-60/J2) (Fig 6C). In addition, three other clonotypes were selected as examples of highly convergent public BCRs (S3 Table). The antibody designated #3978 was a part of the pubCDR3-2 group from D#030, who

had the most PBs belonging to pubCDR3-2 (IGHV3-9/D2-15/J6/IGLV1-40/J3). #5589 and #7665 antibodies were isolated from D#113 and D#008 (both IGHV3-7/D4-17/J6/IGLV3-21/J1). Both participants had public antibodies belonging to the pubCDR3-4 group that each had independently undergone clonotype expansion.

## Binding characterization of public and private IgG antibodies

Following the expression of public antibodies #3978 (pubCDR3-2), #5589 (pubCDR3-4) and #7665 (pubCDR3-4), and private antibody #1664, their binding profile was evaluated against a panel of recombinant HA proteins from current and historical influenza vaccine strains. The mAb #1664 had specificity for H3 HA strains, binding to almost all the rHA proteins from viruses isolated between 1999–2016 influenza seasons. A lower, but still detectable binding was also observed against rHA proteins representing the viruses Switz/17 and S.Aus/19. Conversely, while it was binding to the full-length HA monomer from WI/05, no binding was detected against the chimeric cH5/3 and cH7/3 rHAs with group 2 stem regions, or against the WI/05 HA1 monomer, suggesting binding to the HA1 and HA2 head portion (Fig 6A).

In contrast, mAbs #5589 and #7665 had a preferential binding for seasonal H1N1 strains (Chile/83, Solomon Islands/06, Brisb/07 for #5589 and TX/91, SI/06, Brisb/07 for #7665) and the pandemic strain CA/09. A lower binding was detected for TX/91 for #5589, and Sing/86 and NC/99 for #7665. Similar to #1664, #5589 and #7665 mAbs were also unable to bind the chimeric stem rHA cH6/1, suggesting HA head recognition. However, while #7765 was able to recognize the HA1 monomer from CA/09, #5589 was not, indicating binding to a region encompassing the HA1 and HA2 head portion of the HA. The mAb #3978 had preferential binding for HA proteins from both the Yamagata and Victoria influenza B virus (IBV) lineages. While this mAb was able to recognize the full-length HA trimer and monomer from MA/12, no binding activity was observed against the HA1 monomer of the same strain, indicating binding to a region encompassing the HA1 and HA2 head portion of HA or the stem (Fig 6A).

## Functional characterization of public and private IgG antibodies

In order to evaluate the neutralizing potential of the selected H1-, H3-, and IBV-specific mAbs, their hemagglutination inhibition (HAI) activity was evaluated against the corresponding 2016–2017 influenza vaccine strains (Fig 6B). None of the H1- (#5589 and #7665) and IBV-specific (#3978) mAbs were endowed with HAI activity, even when tested at the maximal concentration of 10 μg/mL against the H1 (CA/09) and IBV (Brisb/08) influenza vaccine strains. Conversely, the H3-specific mAb #1664 was endowed with potent HAI activity against the corresponding vaccine strain, as it was able to inhibit the hemagglutination of the H3 HK/14 virus at a concentration of 0.156 μg/mL.

## Discussion

Influenza virus vaccination remains the most effective way to prevent seasonal influenza virus infection and transmission. The B cell immune responses following vaccination with the 2016–2017 quadrivalent inactivated influenza virus vaccine in 18 to 65 year old participants were previously characterized, and a significantly subdominant antibody response to the H3N2 influenza vaccine component was reported [20]. The 2016–2017 seasonal influenza virus vaccine contained the H1N1 CA/09, the IBV Victoria lineage Brisb/08 and the IBV Yamagata lineage Phu/13 strains that circulated in the human population for the proceeding years. The H3N2 component, the HK/14 vaccine strain, was introduced to replace the previous year's H3N2 strain, Switz/13 in the influenza virus vaccine during the 2016–2017 season. In

this report, a subset of 17 participants that had high serological changes and prominent PB expansion post-vaccination were selected in order to deeply characterize their PB response to influenza virus vaccination through single cell PB BCR sequencing. In accordance with previous reports following hepatitis B [40, 41] and influenza infections [42–45], IgG1 was the most common IgG subclass identified to make an egress from the germinal center in the vaccinated participants in this study (Fig 2A). As for heavy chain variable segment usage, we have seen a significantly higher usage of $V_H3$ than $V_H1$ and $V_H4$ compared to the memory B cell compartment of healthy unvaccinated volunteers, suggesting a preferential representation of vaccine-elicited variable segments (Fig 2D, 2E) [46]. Similarly, the frequency of heavy chain $V_H1$ subfamily usage is disparate from previous reports in healthy unvaccinated participants [8]. Amongst the specific IgG-expressing compartments analyzed for PBs, we observed a marked enrichment of IGHV1-69 heavy chain gene usage (n = 802), particularly when compared with IGHV1-18 (n = 341), which is the predominant $V_H1$ subfamily in unvaccinated participants [46]. This finding is particularly interesting in light of the numerous reports of influenza-specific antibodies belonging to the IGHV1-69 subfamily with heterosubtypic neutralizing activity limited to group 1 viruses, though only a quarter of them carry the motif associated with stem-binding neutralizing IGHV1-69 PBs [9, 46–53]. $V_H3$ and $V_H4$ subfamily usage also showed major discrepancies with those previously reported in unvaccinated participants, suggesting vaccine-related shaping of the BCR response [46]. The subfamilies IGHV3-7 (n = 533) and IGHV4-39 (n = 496), which have been previously shown to be expanded following influenza vaccination, were also highly enriched in our cohort's IgG compartment [9, 46].

Despite extensive studies characterizing the polyclonal antibody responses following influenza vaccination and infection [54, 55], only recent developments in high throughput single cell sequencing allowed for a detailed characterization of PB BCR repertoires at the mAb level with paired heavy-light chain information. In healthy unvaccinated participants, antibodies with a κ light chain are two times more prevalent in the serum than antibodies with a λ light chain (Sigma-Aldrich normal serum reference table). Furthermore, the distribution of IgG ASCs also reflects this disparity with ≅60% κ chain ASC compared to ≅40% λ chain ASC in the periphery, and has been reported to remain largely unaltered following pneumococcal vaccination [56]. In contrast, we observed a bias towards λ light chain usage following vaccination (Fig 2C). This finding is in accordance with a previously published work looking at the distribution of κ/λ light chain ratio in ASCs in the peripheral blood following influenza vaccination [57]. More recently, a similar λ-bias in vaccine positive IgA PBs has been reported [10], which appears highly antigen dependent [58]. Similarly, as previously reported by our group, ferrets first exposed to influenza virus infection also generated a primarily λ-biased antibody response [59]. The three dominant heavy chain variable genes, $V_H1$, $V_H3$, and $V_H4$ showed no significant divergence from the overall κ:λ pairing ratio (40:60). In contrast, $V_H2$ and $V_H7$ had an underrepresentation of κ chains, while $V_H6$ had an overrepresentation of κ chains. However, this may be due to the comparatively low number of PBs recovered that express these $V_H$ gene families ($n_{VH6}$ = 24, $n_{VH7}$ = 15). Acute PB response to influenza vaccination is largely dominated by λ7, with a smaller contribution of λ1 subclass antibodies. Unlike heavy chain subclasses, the intrinsic properties of different light chain subclasses remain elusive, but future studies will assess if this distribution is reflective of the natural B cell repertoire or specific to influenza vaccination response.

Influenza virus vaccination stimulates pre-existing memory B cells, but the impact on *de novo* B cell memory is still controversial [60–62]. The response to influenza virus vaccination is largely oligoclonal and polarized with the largest 6% of the most abundant clonotypes representing up to 60% of the entire repertoire [3]. Furthermore, after influenza virus vaccination ≅60% of the overall serological repertoire results from pre-existing clonotypes [3].

Nonetheless, in young participants, influenza virus vaccination induces somatic hypermutation of pre-existing clonotypes and adaptation to the vaccine strains [62]. In this study, a highly oligoclonal response was observed in the acute PB response following influenza virus vaccination. The proportion of unique clonotypes was not age-dependent ($p$ = 0.054) (S4 Fig) and there was no relationship between a more diverse PB BCR repertoire, as determined by the proportion of unique clonotypes, and a stronger immune response, as measured by ELISA titer ratios and HAI titers to the H1 and the B-Yamagata components [20]. In contrast, a more diverse PB BCR repertoire was negatively correlated with a stronger immune response to the H3 and the B-Victoria vaccine components. Therefore, it appears beneficial to have a few highly expanded antibodies, rather than a large variety of diverse antibodies against the H3 and B-Victoria HA proteins. This is supported by the expression and functional characterization of the selected public and private PBs that featured a cross-reactive profile against historical influenza virus vaccine strains.

For this study, vaccine antigen-specific PBs were not specifically selected by flow cytometry. However, the PB compartment has been shown to be particularly enriched in antigen-specific B cells following infection or vaccination [16, 63]. Additionally, participants were selected based on high and broad serological responses after vaccination and had ~10% of PBs defined as vaccine-reactive cells (Fig 1C). Nonetheless, one cannot exclude the possibility that some PBs secrete self-reactive antibodies or that their cognate antigen is either unrelated to the vaccination or related to other components of the vaccine (*e.g.* influenza NA, M1, M2 or egg proteins), which also elicit antibody responses [64–66]. Future studies will focus on the expression and functional description of the major clonotypes. It will be necessary to validate these clones as influenza virus-specific and define their antigenic and functional activity.

This study focused on categorizing and cataloguing clonally expanded lineages, but PBs that may have undergone extensive proliferative expansion but showed no evidence of clonal expansion were ignored. B cell evolution is critical in providing useful immune reactions to an immediate challenge, in our case vaccination, and heavily affinity matured PBs may only be represented by a single or very few individual PB cells. Nevertheless, they may be a crucial component in the immune response elicited to the vaccine antigens. Moreover, the origin of the sequenced PBs may be discrete, as some may have arisen from naïve B cells in response to vaccination, while others may have been recalled from the memory B cell compartment. A future study will focus on a distinctive look at proliferative PB expansion and the origins of individual PB cells.

Perhaps the most significant finding of this study was the detection of antibody convergent evolution following influenza virus vaccination (Figs 4 and 5). Previous studies have hinted at the possibility of convergent evolution of the B cell repertoire during influenza virus infection or vaccination [67–69] using animal models or by comparing small participant subsets [70, 71]. Here, in a cohort of 17 participants, the acute BCR repertoire of more than 50% of the participants showed signs of convergent evolution, which indicates that an environmental stimulus caused highly similar immune reactions independently in multiple participants. As the participants vary in gender, age, race, and co-morbidities, the influenza virus vaccine they received is the common factor that preceded sample collection and BCR repertoire analysis. Thus, these public antibodies were most likely produced convergently in response to the administration of the influenza virus vaccine.

Convergence group pubCDR3-1, the only convergent group whose heavy and light chain pairing showed high levels of pairing promiscuity. Pairing promiscuity occurs when heavy chains have undergone convergence between different participants, but they pair with strikingly different light chains. Members of group pubCDR3-1 had the most divergence amongst them, with the lowest average percent identity in both full-length V(D)J and CDR3 identity.

Similar heavy chains paired with vastly different light chains, due to heavy chain sequences that likely originated from the same germline sequence pairing with light chains that were distantly related from one another.

Interestingly, the pairing of similar heavy chain with different light chain gene segments suggest a minor contribution of the latter to antigen binding. In fact, most of the resolved co-crystal structures between mAb Fabs and HA antigens revealed minimal interaction between the VL fragment and the HA protein while showing a predominant contribution of the VH fragment [48, 49, 72, 73]. However, even if not directly involved in antigen binding, light chains could modulate the antibody functional activity by affecting the local conformation of heavy chains [74]. In this regard, investigating the role of different light chains on the overall antibody conformation and ultimately on its function is of pivotal importance not only to develop broadly protective mAbs against influenza A viruses and other emerging viruses, but also to shed light on the different mechanisms of antigen/epitope recognition.

Group pubCDR3-2 PBs have the longest HCDR3 peptide lengths, but even though these HCDR3s have possibly undergone several rounds of SHM, increasing the distance from the germline sequence with each consequent round, there is still up to 96% identity between HCDR3 peptides among the PBs in this group from the four different participants. Group pubCDR3-3 had the shortest HCDR3 length and was likely the least affinity matured of the 4 convergent groups. Group pubCDR3-4 is unique in that PBs belonging to this convergent group were only observed in two participants, but clonotype expansion was observed in both.

While the H1-specific public and H3-specific private antibodies feature a subtype-specific, cross-reactive binding profile against historical vaccine strains, they also show decreased binding against more recent "future" strains that were not included in the vaccine formulation at the time of vaccination (2016–2017 season). Interestingly, the selected public H1-specific mAbs were endowed with a broad binding profile against seasonal and pandemic H1N1 strains, suggesting the recognition of a discrete, conserved epitope. However, mAb #5589 did not recognize the chimeric cH6/1 rHA or the HA1 HA portion, suggesting the possible recognition of a region contained in the HA1-HA2 HA interface instead. mAb #7665 is able to recognize the HA1 HA portion, but not able to recognize cH6/1, most likely binding the HA head portion. Additionally, none of the H1-specific antibodies exhibited HAI activity which is further confirmed by similar binding affinity to the Y98F HA mutant.

The selected public IBV-specific mAb #3978 is endowed with a pan-IBV binding profile against all the tested IBV rHA strains belonging to both the Yamagata and the Victoria lineages. This mAb could possibly recognize a region encompassing the conserved HA stem region, as it is not able to recognize the HA1 HA portion, and lacks HAI activity. The lack of HAI activity of the three public antibodies suggests the absence of a protective activity. However, these antibodies can nevertheless be endowed with extra-neutralizing properties, such as the ability to elicit an ADCC- or ADP-mediated activity and thus possibly still confer protection, as already descried for HA1-HA2 interface directed antibodies [3].

Interestingly, the heavy chain variable segment of the private mAb #1664 belonged to IGHV1-69-2. Antibodies with IGHV1-69 have been extensively described as being represented mainly in broadly HA stem-directed group 1 and/or group 2 cross-reactive antibodies, especially when endowed with a motif comprised of 3 amino acid substitutions encompassing the CDR2 and CDR3 regions, described by Avnir and colleagues [49, 52, 53, 75]. Amongst our participants, 24.8% of all the sequenced IGHV1-69 PBs possessed the motif, suggesting they encode for HA stem-directed neutralizing antibodies. Some of our participants had no IGHV1-69 PBs bearing the motif (D#030, D#070, D#113, D#118), while others have the majority of their IGHV1-69 PBs exhibiting the motif (76.8% in D#122, 75% in D#085, 59.1% in D#102). Much less is known about antibodies classified as IGHV1-69-2 and their binding

profiles. None of the 209 IGHV1-69-2 PBs in our study carried the motif, and mAb #1664 was a head-directed antibody, endowed with strong HAI activity but no detectable binding activity when tested against the chimeric cH5/3 and cH7/3 rHAs or the HA1 monomer, suggesting the recognition of a conformational epitope encompassing the HA trimer. Moreover, #1664 exhibited not only a broadly binding profile against current and past H3N2 strains but also showed potent HAI activity at a nanomolar range (~1 nM) against the HK/14 vaccine strain, suggesting the antibody has a potent protective effect. This can partially explain the high HAI activity shown by the corresponding participant's serum against past and current H3N2 vaccine strains [20]. Future studies will be aimed at defining in depth the epitope recognized by #1664 as well as its molecular mechanism of neutralization.

In conclusion, our findings indicate that influenza virus vaccination elicits a unique set of PBs unlike those in unvaccinated humans. This is largely due to differential heavy-light chain pairing and kappa/lambda light chain usage. The most expanded clonotype in some participants can be as much as ~22% of their total PB repertoire. More than 50% of the participants had public PBs that could be classified into convergent groups. While the binding profile of the selected public antibodies was influenza-specific in every case, the lack of neutralization potential in the selected convergent mAbs further confirms the limit of current influenza vaccines to elicit cross-reactive antibodies that are able to bind future circulating influenza viral strains. The subtype-specific nature of the characterized public mAbs confirms that heterosubtypic mAbs are rare and that subtype-specific next generation vaccines (i.e. COBRA approach) can be more effective in recalling and thus eliciting an effective humoral response in a "universal" influenza pre-immune setting. Thus, the need for broadly neutralizing next generation influenza virus vaccines, eliciting cross-reactive anti-influenza virus antibodies are paramount in order to confer a more durable and broadly protective immune response.

## Supporting information

**S1 Graphical abstract. Experimental design.** PBMCs were collected 7 days after influenza virus vaccination. Sorted PBs were subjected to the Immune Repertoire Capture™. Cells were barcoded to identify paired heavy and light chain sequences that were used in the downstream BCR repertoire analyses.
(TIFF)

**S1 Fig. Serological HAI activity landscape.** Serum samples from 17 subjects with significant serological response to influenza vaccine were tested for HAI activity against a broad panel of (A) H1N1 IAV strains from 1918 to 2009, (B) H3N2 IAV strains from 1968 to 2016, (C) IBV Yamagata lineage virus from 1988 to 2013, and (D) IBV Victoria lineage from 1987 to 2017.
(TIFF)

**S2 Fig. Antibody titers.** (A) Serological antibody responses to influenza vaccination in the 17 participants analyzed in this study. (B) Memory B cell-derived antibody responses to influenza vaccination in the 17 participants analyzed in this study.
(TIFF)

**S3 Fig. Differences in variable gene segment usage between participants and by participant age.** (A) Divergence from the mean heavy chain variable segment usage for each participant. (B) Divergence from the mean heavy chain variable segment usage by participant age. (C) Divergence from the mean light chain variable segment usage for each participant. (D) Divergence from the mean light chain variable segment usage by participant age.
(TIFF)

**S4 Fig. Proportion of unique clonotypes is not significantly increased in younger partici-pants ($p = 0.054$).**
(TIFF)

**S5 Fig. Preferential V(D)J segment joining of heavy chain sequences from 17 influenza vac-cinated individuals.** (A) Heatmap of preferential association between V and J gene segments. (B) Heatmap of preferential association between J and D gene segments. (C) Heatmap of pref-erential association between V and D gene segments. Yellow corresponds to low pairing fre-quency, while dark blue corresponds to the highest observed pairing frequency.
(TIFF)

**S6 Fig. Preferential VJ segment joining of light chain sequences from 17 influenza vacci-nated individuals.** (A) Heatmap of preferential association between V and J gene segments for lambda light chains. (B) Heatmap of preferential association between V and J gene segments or kappa light chains. Yellow corresponds to low pairing frequency, while dark blue corre-sponds to the highest observed pairing frequency.
(TIFF)

**S7 Fig. CDR3 characteristics of heavy and light chain amino acid sequences.** (A) HCDR3 length distribution and amino acid frequencies. (B) LCDR3 length distribution and amino acid frequencies for lambda light chains. (C) LCDR3 length distribution and amino acid fre-quencies for kappa light chains.
(TIFF)

**S8 Fig. Percent heavy chain identity of public PBs that are a part of one of the four pubCDR3 convergent groups.** Public PBs are contrasted with non-public PBs belonging to the same heavy chain variable segment, based on (A) heavy chain variable segment nucleotide identity, and (B) HCDR3 peptide identity.
(TIFF)

**S1 Table. PB response to influenza vaccination.**
(DOCX)

**S2 Table. List of the three most expanded clonotypes for each donor.** Colors correspond to those use in Fig 3.
(DOCX)

**S3 Table. List of convergent (public) clonotypes for each donor.** Colors correspond to those use in Fig 5.
(DOCX)

**S4 Table. Key resources.**
(DOCX)

**S1 Text. Strain abbreviations.**
(DOCX)

## Acknowledgments

The authors would like to thank James Allen, Michael Carlock, Naoko Uno, and Bradford Lefoley for technical assistance, as well as the Center for Vaccine and Immunology protein production core, Jeffrey Ecker, Spencer Pierce, and Ethan Cooper for the expression and puri-fication of the recombinant proteins. We also thank Brad Phillips and Kim Schmitz, as well as all of the members of the UGA Clinical and Translational Research Unit (CTRU), and give a

special thanks and appreciation to the volunteer participants in the study. This study was supported in part by computing resources and technical expertise from the Georgia Advanced Computing Resource Center (GACRC), a partnership between the University of Georgia's Office of the Vice President for Research and Office of the Vice President for Information Technology. The content is solely the responsibility of the authors and does not necessarily represent the official views of the NIH. The Graphical abstract was created using Biorender.

## Author Contributions

**Conceptualization:** David Forgacs, Rodrigo B. Abreu, Greg A. Kirchenbaum, Ted M. Ross.

**Formal analysis:** Rodrigo B. Abreu, Giuseppe A. Sautto, Ted M. Ross.

**Funding acquisition:** Ted M. Ross.

**Investigation:** David Forgacs, Elliott Drabek, Kevin S. Williamson, Dongkyoon Kim, Daniel E. Emerling.

**Methodology:** David Forgacs, Rodrigo B. Abreu, Elliott Drabek, Kevin S. Williamson, Dongkyoon Kim, Daniel E. Emerling.

**Project administration:** David Forgacs, Rodrigo B. Abreu, Ted M. Ross.

**Supervision:** Ted M. Ross.

**Writing – original draft:** David Forgacs, Rodrigo B. Abreu.

**Writing – review & editing:** David Forgacs, Rodrigo B. Abreu, Giuseppe A. Sautto, Greg A. Kirchenbaum, Daniel E. Emerling, Ted M. Ross.

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
