## [Decision Letter · Decision Letter 0]

23 Dec 2020

PONE-D-20-32620

Convergent antibody evolution and clonotype expansion following influenza virus vaccination

PLOS ONE

Dear Dr. Ross,

Thank you for submitting your manuscript to PLOS ONE. After careful consideration, we feel that it has merit but does not fully meet PLOS ONE’s publication criteria as it currently stands. Therefore, we invite you to submit a revised version of the manuscript that addresses the points raised during the review process.

During the revision process, please address the concerns related to the analysis of public clonotype data and the availability of all underlying data related to the study.

s

We look forward to receiving your revised manuscript.

Kind regards,

Victor C Huber

Academic Editor

PLOS ONE

Journal Requirements:

2.Thank you for stating the following in the Financial Disclosure section:

'This study was supported in part by resources and technical expertise from the Georgia Advanced Computing Resource Center (GACRC), a partnership between the University of Georgia’s Office of the Vice President for Research and Office of the Vice President for Information Technology. The content is solely the responsibility of the authors and does not necessarily represent the official views of the NIH. '

We note that one or more of the authors are employed by a commercial company: Atreca, Inc.

Reviewers' comments:

Reviewer's Responses to Questions

**Comments to the Author**

1. Is the manuscript technically sound, and do the data support the conclusions?

Reviewer #1: Yes

Reviewer #2: Yes

2. Has the statistical analysis been performed appropriately and rigorously? 

Reviewer #1: Yes

Reviewer #2: Yes

3. Have the authors made all data underlying the findings in their manuscript fully available?

Reviewer #1: No

Reviewer #2: Yes

4. Is the manuscript presented in an intelligible fashion and written in standard English?

Reviewer #1: Yes

Reviewer #2: Yes

5. Review Comments to the Author

Reviewer #1: This study by Forgacs et al. sequenced the B cell receptor repertoire of 17 participants after seasonal influenza virus vaccination. While largely descriptive, the study comprises an interesting data set that could be of interest to the field of influenza research. To allow other researchers to further study this data set, the authors should make the underlying raw data available and accessible.

Minor:

- Please show error bars for the HAI data in Figure 1B

- Please provide specific percentages for IgG2 and IgG3.

Reviewer #2: The authors present an analysis of BCR sequences following influenza vaccination, determined by B cell cloning of approximately 7,800 B cells from PBMCs. The main finding here is evidence of convergent responses, deduced from public VH genes in 10/17 donors analyzed. Overall there is a wealth of data here of interest to the field. A number of issues need to be addressed however, before the manuscript is accepted.

Major:

1) Public clonotypes were defined as ...."PBs whose heavy chain variable region nucleotide pairwise identity was over 80%, and HCDR3 peptide sequence pairwise identity was over 75% compared to at least one other PB from a different participant in the same group". This is very permissive. What is the justification for such a low threshold for HCDR3 aa identity? Did the clonotypes considered as "public" share the same VJ genes. Analysis of public clonotypes with more stringent criteria, ie same V same J and >85% aa identity needs to be performed and reported.

2) Given that public VH were paired with different VL was there any analysis of Mabs to show that the binding specificity was primarily VH driven? Not an essential point, but at least should be discussed?

2) The text is excessively long, especially the Discussion. The text should be reduced by at least 30%.

6. PLOS authors have the option to publish the peer review history of their article (what does this mean?). If published, this will include your full peer review and any attached files.

Reviewer #1: No

Reviewer #2: No

---

## [Author Response · Author response to Decision Letter 0]

30 Jan 2021

Athens, GA - January 29, 2021

Dear Editor: 

We are resubmitting a revised version of our manuscript ID PONE-D-20-32620 entitled “Convergent antibody evolution and clonotype expansion following influenza virus vaccination” to PLoS ONE after having addressed the revisions that were suggested by you and the reviewers. Below, please find a point-by-point reply to the referee comments.

Editor comments:

1. We would like to update the Financial Disclosure section to state the following:

This study was supported by the NIH contract HHSN272201400004C (NIAID Centers of Excellence for Influenza Research and Surveillance, CEIRS). The content is solely the responsibility of the authors and does not necessarily represent the official views of the NIH. The funder provided support in the form of salaries for authors [DF, RBA, GAS, GAK], but did not have any additional role in the study design, data collection and analysis, decision to publish, or preparation of the manuscript. The specific roles of these authors are articulated in the ‘Author contributions’ section. Some of the authors are affiliated with Atreca, Inc., however, they provided no funding for the study, only assisted with the data collection and the preparation of the manuscript.

(Please note also that in the previous submission, GACRC was erroneously included in the Financial Disclosure section. While they provided assistance, it was all technical and not financial, so the statement belongs in the Acknowledgements but not in this section.)

2. In accordance to your request, we would also like to update the Competing Interests Statements to report the following:

Some of the authors are affiliated with Atreca, Inc., but the company provided no funding for the study and no competing interests exist. The commercial affiliation of those authors does not alter our adherence to PLoS ONE policies on sharing data and materials.

3. The statements including the phrase “data not shown” have been replaced by the appropriate references including the data or relating to the statement (lines 534 and 631).

Reviewer comments: 

Reviewer #1 (Response to Questions):

To the question “Have the authors made all data underlying the findings in their manuscript fully available?”, Reviewer #1 answered no. This was correct at the time of the review as only placeholders were left in the manuscript while NCBI authenticated and released the 15,554 novel nucleotide sequences provided by our study. They are now all publicly available in GenBank as a Target Locus Study under accession numbers KEOV00000000 and KEOU00000000. References to those accession numbers have been included in the manuscript (line 190).

Reviewer #1 (Comments to the Author):

1. Please show error bars for the HAI data in Figure 1B

We thank the reviewer for having noticed that and we apologize for our oversight. Error bars have now been added to the HAI panel in Figure 1B, along with the legend now stating that all samples were run in triplicate (line 265).

2. Please provide specific percentages for IgG2 and IgG3.

Specific percentages have been provided for IgG2 and IgG3 in lines 271-272.

Reviewer #2 (Comments to the Author):

1. Public clonotypes were defined as ...."PBs whose heavy chain variable region nucleotide pairwise identity was over 80%, and HCDR3 peptide sequence pairwise identity was over 75% compared to at least one other PB from a different participant in the same group". This is very permissive. What is the justification for such a low threshold for HCDR3 aa identity? Did the clonotypes considered as "public" share the same VJ genes. Analysis of public clonotypes with more stringent criteria, ie same V same J and >85% aa identity needs to be performed and reported.

Public convergent clonotypes are defined by three criteria, all of which had to be true in order to be considered part of the same public clonotype:

1. As Reviewer #2 stated, all B cell belonging to the same clonotype must have had identical heavy chain V and J genes. The text has been appended to include that distinction. 

2. The minimum heavy chain nucleotide percent identities between two B cell had to be a minimum of 80%. 

3. And in addition to strengthen our criteria, the HCDR3 amino acid identity between donors was set at a minimum of 75%.

These criteria are consistent with what is recently published in the literature (e.g. Galson et al., 2020. Front. Imm.; Setliff et al., 2018. Cell Host Microbe). Increasing this cutoff to the recommended 85% would only exclude two of the 98 B cells that we found to be convergent: one from pubCDR3-1 that otherwise has an 83% heavy chain nucleotide identity, the other from pubCDR3-4 which has a staggering 93% heavy chain nucleotide identity. As both of these B cells have the same V and J genes as other members of their convergent clonotypes, as well as heavy chain nucleotide identities exceeding the published norm, we still feel strongly about including them in the analysis. However, we have included a clear description and rationale in the appropriate section (lines 230-234).

2. Given that public VH were paired with different VL was there any analysis of Mabs to show that the binding specificity was primarily VH driven? Not an essential point, but at least should be discussed?

We agree that this point should be touched on, and though we have not analyzed VH/VL specificity for the purposes of this manuscript, we included a paragraph to address this point (lines 578-587).

3. The text is excessively long, especially the Discussion. The text should be reduced by at least 30%.

We have done our best to shorten the Introduction, Materials and methods, as well as the Discussion sections. Though we understand that the manuscript is long, we do believe it to be free of superfluous details. We truly appreciate the lack of strict word limits for PLoS ONE publications.

Sincerely,

Ted M. Ross, PhD

GRA Eminent Scholar in Infectious Diseases

Director - Center for Vaccines and Immunology

Professor - Department of Infectious Diseases

University of Georgia

---

## [Editor Report · Decision Letter 1]

4 Feb 2021

Convergent antibody evolution and clonotype expansion following influenza virus vaccination

PONE-D-20-32620R1

Dear Dr. Ross,

We’re pleased to inform you that your manuscript has been judged scientifically suitable for publication and will be formally accepted for publication once it meets all outstanding technical requirements.

Kind regards,

Victor C Huber

Academic Editor

PLOS ONE
---

## [Editor Report · Acceptance letter]

11 Feb 2021

PONE-D-20-32620R1 

Convergent antibody evolution and clonotype expansion following influenza virus vaccination 

Dear Dr. Ross:

I'm pleased to inform you that your manuscript has been deemed suitable for publication in PLOS ONE. Congratulations! Your manuscript is now with our production department. 

Kind regards, 

on behalf of

Dr. Victor C Huber 

Academic Editor

PLOS ONE